# Future Time Perspective, Priority of Social Goals, and Friend Networks in Old Age: Evidence for Socioemotional Selectivity Theory Using Subjective Age Gap

**DOI:** 10.3390/healthcare11010022

**Published:** 2022-12-22

**Authors:** Moonjo Bae, Sesong Jeon, Katie Walker

**Affiliations:** 1Major in Child & Family Studies, School of Child Studies, College of Human Ecology, Kyungpook National University, Daegu 41566, Republic of Korea; 2Children’s Studies, Child Life and Health Program, School of Social Work, Eastern Washington University, Cheney, WA 99004, USA

**Keywords:** socioemotional selectivity theory, subjective age gap, future time perspective, friend networks, priority of social goal

## Abstract

Several studies have focused on population aging, with a focus on the relationship between age and the main concepts of the socioemotional selectivity theory, but many do not report consistent results. Therefore, this study sought to better understand how the socioemotional selective theory informs our understanding of the elderly in Korea. More specifically, it aimed at observing how age groups differ in regards to future time perspective, social goals, and friend networks. Data were collected from 271 elderly people (M = 72.98 years old, SD = 5.63) using questionnaires. The statistical program SPSS 25.0 was used to perform descriptive statistical analyses, reliability analyses, and ANOVAs. The findings indicated that the Korean elderly participants perceived their subjective age to be younger than their chronological age. Furthermore, if they perceived their subjective age to be older than their chronological age, they were more likely to report that their network of friends was smaller than they desired. Lastly, depending on their age, the Korean elderly participants reported different priorities of the goals they wished to pursue. These results could help researchers, clinical practitioners, and policymakers to better understand the unique differences in the Korean elderly.

## 1. Introduction

Worldwide, we are currently living in an aging society. By 2030, one in six people will likely be 60 or older, and by 2050, the world’s 60-plus population is expected to double (i.e., 2.1 billion) [1]. Between 2020 and 2050, 426 million people will likely be 80 or older [1]. Thus, the need for research focusing on social networking in later life is increasingly important [2,3]. 

According to the socioemotional selectivity theory (SST), the social environment of close friendship networks becomes increasingly important for the elderly population. It is considered that as humans get older, the importance of choice in information pursuit and emotional regulation processes tilts toward emotions [4]. Changes in social motivation also occur as individuals age due to their changing perspective on time, which, in turn, affects the social goals or choices that they pursue. The concepts of “age” and “future time” are the two most common ideas in the SST and were used in this study.

According to the SST, “age” is generally referred to as “chronological age” (e.g., [5,6]), an individual’s actual age, while “subjective age” is a person’s sense of how old they feel, regardless of their actual age. In other words, subjective age is a multidimensional characteristic indicating which age group one categorizes oneself in and includes more social, psychological, and personal meaning than chronological age [7,8]. Pinquart and colleagues (2021) performed a meta-analysis of 294 papers to look at how the difference between subjective and chronological age changed over the course of a person’s life and to assess if the size of this difference was different globally [9]. The results showed that, on average, the elderly felt 10 to 21 years younger than their chronological age. Furthermore, children felt approximately 3 years or 34% older than their real age, whereas older individuals (over 60) felt between 10.74 and 21.07 years, or from 13% to 18%, younger than their chronological age. Additionally, this study found that in all the continents, there was an increase in the difference between a person’s subjective age and their chronological age in adulthood. These findings demonstrated the need to include “subjective age” when studying the elderly, as most do not identify with their chronological age. Additionally, some studies have found that the older one’s chronological age is, the bigger the disparity with one’s subjective age is [10]. For example, Shinan-Altman and Werner [11] found that the difference between the perceived and desired age was significantly greater in the elderly than in the middle aged. 

In general, subjective age is often measured with a single question, “how old do you feel?”. Kleinspehn-Ammerlahn et al. [12] introduced appearance age (“how old do you feel when you look at yourself in a mirror?”). In this study, in total, four types of subjective age were examined together by adding Stephens’ [13] interest and activity age to reflect a scale that can measure subjective age more multidimensionally. Furthermore, the difference score between chronological and subjective age was used. We acknowledge that using scores based on differences can be controversial [14]. However, when using difference scores, one can directly determine how much younger or older an individual perceives themselves to be (e.g., 3 years younger). In addition, the partial calculation of chronological age using subjective age may lead to the disappearance of age-related variances [12]. This supports the argument that the use of inconsistent scores should be considered to control the various effects of chronological age [15]. Therefore, this study attempted to examine the subjective age gap.

According to the SST, future time perspective is an important concept that explains which social goals people choose and refers to the subjective perception of how much time is left for them in the future, rather than the physical form of time expected to be equal for everyone [16]. According to the SST, the pursuit of prioritized social goals fluctuates depending on future temporal constraints. For example, in younger people, goals, such as knowledge seeking, are prioritized, whereas in the elderly, meaningful and happy experiences are prioritized [4]. According to the SST, an individual with an open-ended future time perspective considers that the time given to them is sufficient, aims to acquire new experiences and knowledge, and pursues knowledge-related goals, such as expanding the breadth of their personal connections. On the other hand, an individual with a limited future time perspective recognizes that there is not much time left for them in the future; pursues emotional goals, such as the acquisition of emotional bonds and belonging to intimate others; has a positive disposition; pursues their meaning in life; and engages in interpersonal relationships with intimate others [17]. Research has found that people who have an extended future time perspective are more honest, more mobile, and more inclined to act in a healthy manner [18,19]. In a study by Lang [20], the elderly were also selective about spending time together. The elderly interacted more closely and for longer with a few close friends or family members with whom they felt emotionally comfortable. It was found that not only the elderly, but also young people tried to interact with individuals with whom they were emotionally close when they felt that their remaining time was limited. However, some studies have found that age has nothing to do with how motivated elderly individuals are to obtain information and control their emotions [21].

The most basic studies examining the SST have been performed by focusing on which social peers people prefer based on their age or future time perspective (e.g., [17,22,23]). For example, various people, such as “the author of a recently read book”, a “friend”, and “family”, were presented to individuals of various ages and they were asked to classify these people in the order of with whom they would prefer to meet from the most to the least [22]. Results indicated that older people tend to prefer to hang out with people they already know, get closer to them, feel emotionally satisfied, and put emotional goals first because they feel like time is running out [17,22]. Furthermore, a study performed by Ji et al. [23] asked the elderly and young adults to write about three important life events every day for 14 days in their daily lives to evaluate their value. As a result, both the elderly and young adults recorded events related to emotional goals and knowledge-acquisition goals, but the number of events related to emotional goals was 2.12-times higher for the elderly than for young adults. However, both the elderly and young adults recorded more emotional-related goals than knowledge acquisition goals, indicating that events related to emotional goals in daily life yielded more attention. In addition, studies have found that the size of the friendship network or social network decreases with age [24], which might result in loneliness or isolation. In a study of 422 women aged 31 to 77, the relationship between friendship network, subjective age, and life satisfaction was related to low subjective age but not to chronological age [19]. It was found that the more frequently you visit your friends, the lower your subjective age and the higher your life satisfaction [25].

The results of the aforementioned studies indicate that additional research needs to be conducted to see if there is a difference in how people think about the future, their friend networks, and their social goals based on how old they feel they are. In this study, the subjective age gap was used to explore SST instead of only using chronological or subjective age, as was done in previous studies. Further, a questionnaire is used instead of an experiment to see how social goal preferences and friend networks are different based on future time perspective.

## 2. Methods

### 2.1. Participants and Procedure

This study recruited subjects who could voluntarily respond to a survey through religious institutions, welfare centers for the elderly, lifelong education centers, and online cafes. A preliminary survey was conducted with 20 elderly people (65–96 years old) from 24 January to 15 February 2020, to determine the respondents’ understanding of the survey questions and the time required to complete. Based on this, the text size and sentence structure were changed to make it easier for the individuals to understand the questions. The participants in this study were chosen using a snowball sampling method. Due to the severity of the COVID-19 pandemic during the time of the survey, it was nearly impossible to meet the participants in person; therefore, it was decided to mail a questionnaire to the participants and have them respond in writing. To guarantee anonymity, when the participants mailed in the survey, an envelope with the address virtually set by the researcher was used in the information column for the sender. In total, 400 surveys distributed for a total of six months from 1 March 2020 to 31 August 2020, 323 were collected. For this analysis, 271 copies were used. Questionnaires with incomplete or missing answers were not used. To avoid tampering with the data, we used the listwise deletion method to deal with missing data, as too many missing independent valuables can result in the absence of any useful insights.

### 2.2. Instruments

This study constructed a questionnaire using the following measurement tools.

#### 2.2.1. Subjective Age Gap

An individuals’ subjective age is the age one perceives themself. The survey used in Stephens [13] was modified and supplemented for this study. The survey consists of four questions: sensory age (I feel I am 00 years old), appearance age (I think I look 00 years old), interest age (I have the same interests and interests as 00 years old), and activity age (I act as if I am 00 years old). To determine the subjective age gap, it is the average of the participants’ subjective age minus their chronological age. For example, if participant A’s sensory age is 70, the appearance age is 68, the interest age is 60, the activity age is 70, and the chronological age is 69, the value of subjective age gap is (70 − 69) + (68 − 69) + (60 − 69) + (70 − 69)/4 = −2.25 years old.

#### 2.2.2. Future Time Perspective (FTP)

The Future Time Perspective means recognition of how much time you think you have left in your life [4]. The FTP scale developed by Lang and Carstensen [4] was used for this study. This scale consists of questions about perception of the remaining time of life, that is, whether one perceives one’s life as limited or expanded, and there are a total of 10 questions corresponding to a 5-point Likert scale. In this study, the survey was converted into a 7-point Likert scale to increase score variability. On a scale from 1 (strongly disagree) to 7 (strongly agree), participants ranked the extent to which they agreed with each of 10 statements. Sample items are “Many opportunities await me in the future”, “I have plenty of time in my life to make new plans“, and “My future seems infinite to me”. The average score from a total of 10 questions was used for this analysis. This was done after three questions of negatively worded items were reversed coded (e.g., “I have the sense that time is running out”, and “As I get older, I begin to experience that time is limited”). The higher the score, the more time is left in the future, and the Cronbach’s value of FTP was 0.89.

#### 2.2.3. Friend Networks

Since friend relationships are formed voluntarily and individuals may have very different ways and standards for defining friends, this study used the “friend network function” scale of Lee and Han [26], who organized the questions around the “function” of friends. Friends’ functions can be largely divided into three categories: first, as an intimate listener who can disclose secrets; second, as a companion who shares daily and leisure activities; and third, as a source of social stimulation (social stimulator). In the study of Lee and Han [26], each function was measured as two questions, and the scale consists of a 5-point scale ranging from 0 point (none) to insufficient (1 point) and sufficient (4 points). Since the score of each function was calculated as the sum of the scores of the two questions, the score range is from 0 to 8. In this study, the average score for each function was used by transforming it into a 5-point Likert scale from 1 point (none) to sufficient (5 points). This scale consists of a total of 6 questions, 2 questions for each area. The higher the score for each area, the more friends I have as listeners who understand each other (the role of “confidant”) and listen to my worries or stories as if they were my work (the role of “companionship”), and I have a lot of friends who grow me up or stimulate my way of thinking (the role of “social stimuli”). An example of each question and the internal consistency are as follows: a function as a confidant (e.g., a friend who can understand each other without saying anything; a friend who listens to me as if it were his own business when I was in trouble, Cronbach’s α = 0.85); the function of companionship (a friend who stays with you when he has time; a friend who can be with you even if he does not have any particular business, Cronbach’s α = 0.82); a function of social stimuli (a friend who grows me up; a friend who stimulates my mindset, Cronbach’s α = 0.75).

#### 2.2.4. Priority of Social Goal

Priority of Social Goal refers to prioritizing how important people are to themselves among the goals and plans they have in life [4]. To measure this, “the priority of goal domains” developed by Lang and Carstensen [4] was used. The subcategories of Priority of Social Goal are social acceptance, autonomy, emotion regulation, and productivity, and each area consists of three items, for a total of 12 questions. It is measured on a 5-point Likert scale that ranges from not important at all (1 point) to very important (5 points), and the higher the score, the more important each area has to be. An example of a question by sub-area and the internal consistency of the question are as follows: social acceptance (e.g., having a good friend who admits to who I am, Cronbach’s α = 0.79); autonomy (e.g., to be able to decide my future for myself, Cronbach’s α = 0.73); emotion regulation (e.g., knowing myself and my feelings well, Cronbach’s α = 0.80); productivity (e.g., helping others find their goals in life, Cronbach’s α= 0.72).

### 2.3. Analytic Plan

For the analysis of the data collected in this study, the SPSS 25.0 statistical program was used to analyze the data as follows, according to the research purpose:

First, a descriptive statistical analysis (e.g., frequency and percentage) was completed to better understand the participants.

Second, the Cronbach’s α was calculated and analyzed to verify the reliability of all measurement tools.

Third, the Scheffë’s post hoc tests with ANOVA (one-way analysis of variance) were used to examine future time perspective, friend networks, and priority of social goal according to the subjective age gap.

## 3. Results

The general characteristics of the participants of this study are shown in Table 1. In terms of chronological age, 50.2% of individuals were between the ages of 70 and 75. The gender ratio was 28.4% for men and 71.6% for women. A total of 66.1% of the elderly said they were happy to some extent, and high school graduation accounted for the most, with 35.1 percent. A total of 67.5% of the respondents said their economic level was average, and 50.9% of the respondents said their health condition was average. To examine the structural characteristics of friend relationships, 44.3% of the respondents said they had between four and seven friends. Friend proximity is defined as how many close friends can visit each other’s house within an hour, with 38.7% of the respondents saying that half of them live close by and half far away. The frequency of contact with close friends was 56% for those who met more than once a week.

A one-way ANOVA and post hoc test were performed to compare the effect of subjective age gap of the elderly on future time perspective, friend networks, and priority of social goals (Table 2).

First, the results revealed that there was a statistically significant difference in mean FTP between at least two groups (F = 5.475, *p* < 0.001). Scheffë’s test for multiple comparisons found that the mean value of FTP perspective was significantly different between older people who perceive themselves as more than six years younger than their chronological age (M = 3.23, SD = 0.87; M = 3.22, SD = 1.14) and older people who perceive themselves as more than 10 years older than their chronological age (M = 2.50, SD = 0.98). That is, older people who perceive themselves as more than six years younger than their chronological age tend to think they have more time in the future than older people who perceive themselves as more than 10 years older than their chronological age.

Next, it was found that there are significant mean differences in friend networks depending on the subjective age gap. The following results were examined by subfactors of friend networks: In the case of the “intimate listener”, it was found that compared to the elderly who perceive themselves as more than 10 years older than their chronological age (M = 2.69, SD = 0.98), the elderly who perceive themselves as even one year younger than their chronological age (M = 3.38, SD = 0.90; M = 3.32, SD = 0.78; M = 3.32, SD = 0.96) think there are enough friends who listen to their stories (F = 5.483, *p* < 0.001). In the case of the “social simulator,” the elderly who perceived themselves as 1–10 years younger than their chronological age (M = 2.96, SD = 0.94; M = 3.02, SD = 0.82) thought that they had more friends who grew up or stimulated their thinking than the elderly who perceived themselves as 10 years older than their chronological age (M = 2.46, SD = 1.04) (F = 3.925, *p* < 0.01). Likewise, in the case of “companion,” the elderly who perceived themselves as 1–10 years younger than their chronological age (M = 3.34, SD = 1.18; M = 3.44, SD = 0.87) felt that I was experiencing a lot more friend experiences where I could spend my free time together when I had time or without any special business than the elderly who perceived themselves as 10 years older than their chronological age (M = 2.81, SD = 1.05) (F = 4.279, *p* < 0.01).

Lastly, there was a statistically significant mean difference in the priority of social goals according to the subjective age gap. The results examined by subfactors of priority of social goals are as follows: In the case of “social acceptance,” it was found that the elderly who perceive themselves even one year less than their chronological age (M = 3.63, SD = 0.66; M = 3.87, SD = 0.65; and M = 3.90, SD = 0.74) prefer social acceptance to the elderly who perceive themselves 10 years or older than their chronological age (M = 3.25, SD = 0.86) (F = 8.859, *p* < 0.001). In the case of, “autonomy,” the elderly who perceived themselves as more than 6 years younger than their chronological age (M = 3.71, SD = 0.54; M = 3.72, SD = 0.71) preferred autonomy goals compared to the elderly who perceived themselves as more than 10 years older than their chronological age (M = 3.06, SD = 0.85) or 1 to 5 years younger than their chronological age (M = 3.31, SD = 0.64) (F = 12.739, *p* < 0.001). Likewise, in the case of “emotion regulation,” the elderly who perceived themselves as more than 6 years younger than their chronological age (M = 3.81, SD = 0.55; M = 3.82, SD = 0.59) preferred emotion regulation goals compared to the elderly who perceived themselves as more than 10 years older than their chronological age (M = 3.23, SD = 0.80) or 1 to 5 years younger than their chronological age (M = 3.44, SD = 0.69) (F = 11.709, *p* < 0.001). In the case of “productivity,” the elderly who perceived themselves as 6–10 years younger than their chronological age (M = 3.44, SD = 0.54) preferred productivity goals compared to the elderly who perceived themselves as more than 10 years older than their chronological age (M = 2.89, SD = 0.87) (F = 7.623, *p* < 0.001).

## 4. Discussion

In this study, three main results will be discussed. First, the elderly perceived their subjective age as younger than their chronological age. This shows that the FTP is occurring more positively in the elderly. Most older people report being younger than their chronological age [27], and the younger they perceive themselves, the more positive they are in old age. Teuscher [28] stated that considering one’s subjective age as younger can buffer negative emotions. Older people who report a lower subjective age than their peers use more beneficial coping strategies and tend to feel younger than their age [29]. Therefore, considering oneself to be younger in subjective age shows having extended FTP. In fact, according to empirical studies dealing with FTP and mental health, it is reported that people with more extended future time concepts have better emotional health [30], better memory and enforcement functions [31], a low risk of cognitive impairment [32], and a high mortality rate [33]. On the other hand, people with limited FTP showed increased depression and decreased optimism and their life expectancy also tended to be short [34,35]. Interestingly, people with “limited” FTP have a lower sense of control on social interaction, which is associated with worse mental-health outcomes. Since the elderly population typically reports higher levels of depression, one way to help the elderly is to mitigate their loss of social networks. In addition, from a motivational point of view, FTP encourages people to be purposeful and self-controlled [36], so it is expected that creating a learning environment will promote extended FTP for the elderly. For example, if the elderly population is informed of investment methods, such as time, plan, effort, and resources, and receives continuous training to apply them to their daily lives, their FTP will expand and their emotional health will also increase.

Second, when observing the friend network according to the subjective age gap of the elderly in this study, they felt that the friend network was insufficient if they perceived their subjective age as greater than their chronological age. It can be inferred that this is the result of a reduction in size of their friendship network, which may result from one feeling subjectively close to death. Lang [37] also found that the elderly involuntarily decreased their relationships due to the deaths of their spouses or friends in their social network. In addition, the study found that participants perceived their friend network as larger if their subjective age was younger than their chronological age [25]. In particular, women felt that they were younger than their actual age and had a larger friendship network and women with many friends experienced higher life satisfaction [25]. Likewise, existing empirical studies have found that feeling older than one’s actual age was associated with increased psychological pain, decreased subjective and objective physical health, and decreased difficulty participating in daily activities [38,39]. The friend networks according to the subjective age gap used in this study also support the existing research results. Therefore, as part of a method to improve the mental health of the elderly through friend networks, several attempts to recognize their subjective age as young will be needed. For example, choose a place to greet your friends and practice steadily in your daily life to have control over your life, such as where to put the flowers to show them and how to treat them.

Third, according to this study, elderly people who perceived themselves as younger than their chronological age reported high “emotion regulation”. In other words, the younger the perceived elderly, the higher the emotion regulation goal, which is a different result from existing studies on SST. This may be because the elderly put less emphasis on information-seeking goals than young adults [37]. This partially supports SST’s claim that the priority of social goals changes with age, but it shows that the elderly may not pursue emotional goals more strongly.

Through these results, suggestions for follow-up research are as follows. Self-reported social motivation among young people suggests that emotion regulation goals are stable throughout adulthood, but information-seeking goals decrease with age. Lang and Carstensen [40] stated that in the area of social goals (i.e., autonomy, social acceptance, productivity, and emotion regulation), when subjects think that there is not much time left in the future, they prioritize productivity goals and goals related to emotion regulation. However, less time left can increase the time pressure to realize important goals, so more research should explore whether the goal of emotional regulation is preferable [41]. Further, research has solidified that people’s preferences for social goals change with age, but more research is needed to better understand prosocial goals for the elderly.

It is important to note that this study has limitations. First, in order to find out the social network of the elderly, only the functional aspects of the friendship network were examined, and it is necessary to look at the larger social network together. Second, the studies of elderly group segmentation will be more useful if there are more seniors who are 76 years old or older. Third, both productive and emotional goals can be valued regardless of age, so it would be better to use a measure to see if the two are compatible. Further, the “subjective age gap” was calculated by taking the average of the difference between the chronological age and the subjective age in the current study. Another possibility would be to use an average of the absolute values of the differences. It might be interesting to see if researchers find different results using an absolute value measure. Fourth, except for “Friend Networks”, we used the original tools translated into Korean. However, depending on the nature of the study being undertaken and the specific instruments employed, the barriers encountered by cross-cultural researchers throughout the translation process may differ. Finally, as confounders, we did not control for sociodemographic variables. The main aim of this study was to analyze the different means within the groups, not to analyze covariance. However, when considering ANCOVA (Analysis of Covariance), which has better power, interaction detection and estimation, and improvements to overcome variable measurement error, it could yield more meaningful outcome means than ANOVA.

Despite these limitations, this study has several significant impacts. First, the subdivision of the older age group is an approach that is limited in the current literature. By verifying whether the elderly group is the same group or not, it will be easier to understand the characteristics and needs of each age group. By breaking up the age group, researchers, clinicians, and policymakers will be able to understand the differences between the very young and the very old within the elderly subpopulation. This information will help individuals create better welfare policies for each subgroup.

Second, as a result of examining whether there is a difference in the chronological age and subjective age of the elderly in terms of the priority of social goals, the results justify that leisure for, education for, and support for the elderly should be provided.

It will be necessary to help the elderly actively determine activities and goals in their lives, enabling them to think more positively about the future.

Lastly, changing attitudes around aging will ultimately enhance the lives of older people.

## 5. Conclusions

This study examined whether there are significant mean differences in future time perspective, friend network, and priority of social goals depending on the subjective age gap of the elderly, not their chronological or subjective ages, which are considered important in SST (Socioemotional Selectivity Theory). According to a survey of 271 elderly Koreans aged 65 or older, the younger the subjective age compared to their chronological age, the more positive the future time perspective. In addition, the elderly who perceived their subjective age to be older than their chronological age felt that their friend networks were insufficient, and their emotion regulation goals were low. These findings are partially consistent with the existing SST research results, but they also show inconsistent results. This can help to understand the characteristics and needs of each more detailed elderly age group. Through this, we will be able to find a way to escape prejudice against the entire elderly population and live in harmony with the elderly in an aging society and prepare for upcoming old age.

## Figures and Tables

**Table 1 healthcare-11-00022-t001:** General characteristics of participants.

	Frequency (N)	Percentage (%)
Age in years		
65~69	62	22.9
70~75	136	50.2
76 and above	73	26.9
Gender		
Male	77	28.4
Female	194	71.6
Happiness level		
Above happy	37	13.7
Average happy	179	66.1
Below happy	53	19.6
Least happy	2	0.7
Education level		
Lower primary school	76	28.0
Middle school	73	26.9
High school	95	35.1
Higher college degree	27	10.0
Economic status		
Very poor	5	1.8
Poor	27	10.0
Average	183	67.5
Good	48	17.7
Very good	8	3.0
Health status		
Very poor	7	2.6
Poor	72	26.6
Average	138	50.9
Good	45	16.6
Very good	9	3.3
Number of friends		
0~3	98	36.2
4~7	120	44.3
Above 8	53	19.6
Number of friends in close proximity		
Very little	22	8.1
Not many	58	21.4
Half close and half far away	105	38.7
Many	50	18.5
A great many	36	13.3
Frequency of contact with close friends		
More than twice a week	76	28.0
Once a week	76	28.0
Once a month	72	26.6
Several times a year	31	11.4
Less than once a year	16	5.9

**Table 2 healthcare-11-00022-t002:** Results of the one-way ANOVA and Scheffë’s post hoc tests.

		More than 10 Years Old(*n* = 39)	Less than 1~5 Years Old(*n* = 82)	Less than 6~10 Years Old(*n* = 63)	Less than 11 Years Old(*n* = 87)	F
Future Time Perspective						5.475 ***
M(SD)		2.50 (0.98)	2.96 (1.04)	3.23 (0.87)	3.22 (1.14)	
Scheffë’s test		a	ab	b	b	
Friend Networks	Intimate Listener					5.483 ***
	M (SD)	2.69 (0.98)	3.38 (0.90)	3.32 (0.78)	3.22 (0.96)	
	Scheffé’s test	a	b	b	b	
	Social Stimulator					3.925 **
	M (SD)	2.46 (1.04)	2.96 (0.94)	3.02 (0.82)	2.74 (0.89)	
	Scheffë’s test	a	b	b	ab	
	Companion					4.279 **
	M (SD)	2.81 (1.05)	3.34 (1.18)	3.44 (0.87)	3.26 (.78)	
	Scheffë’s test	a	b	b	ab	
Priority of Social Goal	Social Acceptance					8.859 ***
	M (SD)	3.25 (0.86)	3.63 (0.66)	3.87 (0.65)	3.90 (0.74)	
	Scheffë’s test	a	b	b	b	
	Autonomy					12.739 ***
	M (SD)	3.06 (0.85)	3.31 (0.64)	3.71 (0.54)	3.72 (0.71)	
	Scheffë’s test	a	a	b	b	
	Emotion Regulation					11.709 ***
	M (SD)	3.23 (0.80)	3.44 (0.69)	3.81 (0.55)	3.82 (0.59)	
	Scheffë’s test	a	a	b	b	
	Productivity					7.623 ***
	M (SD)	2.89 (0.87)	3.02 (0.64)	3.44 (0.54)	3.31 (0.80)	
	Scheffë’s test	a	ab	c	bc	

** *p* < 0.01, *** *p* < 0.001 Note. Scheffë’s test results a, b, and c indicate significantly different groups of means (*p* < 0.05). In other words, means with the same letter are not significantly different. In the case of the “ab” group, it means a group that is vaguely sandwiched between the two, which is not much different from group “a” and not different from group “b”. Note. This study was not collected by limiting the age of the elderly. However, as a result of collecting actual data and classifying them by chronological age, the number of super-aged people aged 80 or older was small, and the number of elderly people in their 70s was relatively large. After looking at the data collected about the difference between chronological and subjective ages that are important to this study, setting them as a cutting point in units of 5 years and treating it as a group of extremes (more than 10 years old, less than 11 years old) was needed to not break the rule for one-way ANOVA analysis (the dependent variable is normally distributed).

## Data Availability

Not applicable. The participants in this study did not give written consent for their data to be shared publicly, so due to the sensitive nature of the research, supporting data are not available.

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
