# Peer review of "Future Time Perspective, Priority of Social Goals, and Friend Networks in Old Age: Evidence for Socioemotional Selectivity Theory Using Subjective Age Gap"

_healthcare, 2022, doi:10.3390/healthcare11010022_

Round 1

Reviewer 1 Report

1.  Lines 14-15  "whether it was" is not clear.  What were you connecting to future time perspective, etc.?

2.  line 17.  I don't think the use of etc. is appropriate scientifically.  I'd prefer to see "and ANOVA."

3.  line 18  I would suggest something like "Second, if they perceived their subjective age as older than their choronological age, they were more likely to report that their network of friends was smaller than they desired."

4.  line 34  I'd suggest "as more important"

5.  line 45  I'd suggest "feel approximately 3 years or 34% older than"

6.  line 52  I'd suggest "The older the chronological age"

7.  line 62  I'd suggest "An open-ended future time perspective"

8.  line 66.  I'd suggest "a limited future time perspective"

9.  line 79  I'd suggest "The most basic checks"

10.  line 119  I'd suggest "who did the preliminary survey"  I am assuming that these 20 are the same as the 20 mentioned at line 115.

11.  line 130   Another possibility would be to use an average of the absolute values of the differences.  I am not saying you need to do this but it might be an interesting thing to see if you get different results using an absolute value measure.

12.  line 213, Table 1.  The results for age in years are not aligned with the proper rows.

13.  line 218.  I'd suggest "Scheffe's test indicate significantly different groups of means (p < .05)"   Or use whatever level of p you used with the Scheffe test. 

14.  line 270  I'd suggest "age as younger can buffer"

15.  The English in this article needs to be reviewed by an English expert as I may have missed some useful improvements.

16.  line 353.  The data availability statement should be more specific.  If the data are not available for some reason, say so and why; if they are available upon request, explain how the request should be done, who is the chief point of contact and their email, address.

17.  Acknowledgements:  please fill in rather than using the stock answer.

18.  I would like to see more about how this paper expands our knowledge of, use of SST, or reveals limitations in SST or raises more questions about SST and what those questions might be.

Author Response

Reviewer Report on Manuscript # healthcare-2073410 “Aging and Mental Health: The Evidence of Socioemotional Selectivity Theory Using Subjective Age Gap”

Note: To facilitate our responses to all the comments, we have numbered each comment and provided the reviewer’s excerpts in italics. Our detailed responses follow in regular font with red color.

First of all, we sincerely appreciate the time and efforts you have provided. Your thoughtful suggestions have helped us substantially improve on our original submission. Please find our detailed response to your comments below.

Reviewer1.

  1. Lines 14-15. "whether it was" is not clear.  What were you connecting to future time perspective, etc.?

Thank you for your comments. To clarify the sentence, we revised the writing. Please see lines 14–16.

  1. line 17.  I don't think the use of etc. is appropriate scientifically.  I'd prefer to see "and ANOVA."

Thank you for your careful comment. Please see line 23.

  1. line 18  I would suggest something like "Second, if they perceived their subjective age as older than their choronological age, they were more likely to report that their network of friends was smaller than they desired."

According to your kind suggestion, we revised the sentence. Please see lines 25-26.

  1. line 34  I'd suggest "as more important"

According to your suggestion, we revised the sentence. Please see lines 42-43.

  1. line 45  I'd suggest "feel approximately 3 years or 34% older than"

According to your suggestion, we revised the sentence. Please see line 62.

  1. line 52  I'd suggest "The older the chronological age"

According to your suggestion, we revised the sentence. Please see line 73.

  1. line 62  I'd suggest "An open-ended future time perspective"

According to your suggestion, we revised the sentence. Please see lines 101-102.

  1. line 66.  I'd suggest "a limited future time perspective"

According to your suggestion, we revised the sentence. Please see line 105.

  1. line 79  I'd suggest "The most basic checks"

According to your suggestion, we revised the sentence. Please see line 120.

  1. line 119  I'd suggest "who did the preliminary survey"  I am assuming that these 20 are the same as the 20 mentioned at line 115.

Thank you for your good point. According to your suggestion, we revised the sentence. Please see line 162.

  1. line 130   Another possibility would be to use an average of the absolute values of the differences.  I am not saying you need to do this but it might be an interesting thing to see if you get different results using an absolute value measure.

Thank you for your thoughtful comments. We added your valuable suggestion in the discussion section. Please see lines 398-401.

  1. line 213, Table 1.  The results for age in years are not aligned with the proper rows.

Thank you for your careful comment. We revised the numbers with the proper rows. Please see Table 1, line 264.

  1. line 218.  I'd suggest "Scheffe's test indicate significantly different groups of means (p < .05)"   Or use whatever level of p you used with the Scheffe test. 

According to your suggestion, we revised the sentence. Please see line 270.

  1. line 270  I'd suggest "age as younger can buffer"

According to your suggestion, we revised the sentence. Please see line 332.

  1. The English in this article needs to be reviewed by an English expert as I may have missed some useful improvements.

Thank you very much for your careful comment. We asked an expert who is fluent in English to review the entire manuscript and revise it by reflecting on it.

  1. line 353.  The data availability statement should be more specific.  If the data are not available for some reason, say so and why; if they are available upon request, explain how the request should be done, who is the chief point of contact and their email, address.

Thank you for your comments. We explained why the data were not available. Please see lines 454-456.

  1. Acknowledgements:  please fill in rather than using the stock answer.

Thank you for your comments. We mistakenly didn’t remove this section. We don’t have any reports for acknowledgements. Please see lines 457-459.

  1. I would like to see more about how this paper expands our knowledge of, use of SST, or reveals limitations in SST or raises more questions about SST and what those questions might be.

Thank you for your valuable suggestions. We revised the discussion section including practical implication of the study and how to use the results for such aims. Please see lines 337-350, 361-371, and 398-401.

Reviewer 2 Report

The research presented in this article concerns an essential ongoing issue. Although the article is socially crucial and interesting, I have some suggestions, pointed out below:

 Title

Aging and Mental Health: The Evidence of Socioemotional Selectivity Theory Using Subjective Age Gap – in my opinion the title is overgeneralization. Moreover, I do not see the mental health perspective in the paper, that justifies that point. The title should be more precisely connected with variables in the study.

 Introduction

The literature review part should be expanded and focused on more relevant new scientific sources. I suggest to include some more information about understanding the subjective age and specificity of late adulthood period. Please consider to explore and include for example the following:

Carstensen, L. L. (2021). Socioemotional selectivity theory: The role of perceived endings in human motivation. The Gerontologist61(8), 1188-1196.

Zadworna, M. (2022). Pathways to healthy aging–Exploring the determinants of self-rated health in older adults. Acta Psychologica, 228, 103651.

Kotter-Grühn, D., Kornadt, A. E., & Stephan, Y. (2016). Looking beyond chronological age: Current knowledge and future directions in the study of subjective age. Gerontology, 62(1), 86-93.

Shinan-Altman, S., & Werner, P. (2019). Subjective age and its correlates among middle-aged and older adults. The International Journal of Aging and Human Development, 88(1), 3-21.

 Methods

 -          Was any missing data in the study? What kind of method was implemented to handle the missing data?

-          Tools – were the Korean versions of the tools used?

-          The subjective age measurement – please explain in more details the chosen way of this measurement, maybe also in the Introduction. Are there another ways of measurement subjective age? The source you cite is Stephens [20],  from 1991. What is the reason for choosing this source and method?

 Data Analysis

        - Did you checked the data distribution normality?

- Did you control for sociodemographic variables as confounders? Please explain

 Results

 -          More detailed information should be given about the age gap at the beginning – the table 2 is not clear. How and why authors introduce the categories: More than 10 years old, Less than 1-5 years old and others

- Above the Table 2: Note. a, b, and c are not clear. The reader may not understand what groups differ from which groups.

 Discussion

 -          What are practical implication of the study? Please include some more information about the mental health promotion for elderly – how to use the results for such aims?

 General suggestion – the paper should be proofread

Author Response

Reviewer Report on Manuscript # healthcare-2073410 “Aging and Mental Health: The Evidence of Socioemotional Selectivity Theory Using Subjective Age Gap”

Note: To facilitate our responses to all the comments, we have numbered each comment and provided the reviewer’s excerpts in italics. Our detailed responses follow in regular font with red color.

First of all, we sincerely appreciate the time and efforts you have provided. Your thoughtful suggestions have helped us substantially improve on our original submission. Please find our detailed response to your comments below.

Reviewer2.

The research presented in this article concerns an essential ongoing issue. Although the article is socially crucial and interesting, I have some suggestions, pointed out below:

 Title

Aging and Mental Health: The Evidence of Socioemotional Selectivity Theory Using Subjective Age Gap – in my opinion the title is overgeneralization. Moreover, I do not see the mental health perspective in the paper, that justifies that point. The title should be more precisely connected with variables in the study.

Thank you so much for your guidance. Following your suggestion, we changed the title to something more precisely connected with the variables in the study. Please refer to lines 2-3. 

 Introduction

The literature review part should be expanded and focused on more relevant new scientific sources. I suggest to include some more information about understanding the subjective age and specificity of late adulthood period. Please consider to explore and include for example the following:

Carstensen, L. L. (2021). Socioemotional selectivity theory: The role of perceived endings in human motivation. The Gerontologist, 61(8), 1188-1196.

Zadworna, M. (2022). Pathways to healthy aging–Exploring the determinants of self-rated health in older adults. Acta Psychologica, 228, 103651.

Kotter-Grühn, D., Kornadt, A. E., & Stephan, Y. (2016). Looking beyond chronological age: Current knowledge and future directions in the study of subjective age. Gerontology, 62(1), 86-93.

Shinan-Altman, S., & Werner, P. (2019). Subjective age and its correlates among middle-aged and older adults. The International Journal of Aging and Human Development, 88(1), 3-21.

Thank you so much for your feedback. We included more information about subjective age after exploring the papers you recommended. Please see lines 55-57 and 73-94.

 Methods

-   Was any missing data in the study? What kind of method was implemented to handle the missing data?

Listwise deletion method was applied to handle missing data. Please see lines 173-175.

-   Tools – were the Korean versions of the tools used?

      Except for "Friend Networks", we used the original scale translated into Korean.

-   The subjective age measurement – please explain in more details the chosen way of this measurement, maybe also in the Introduction. Are there another ways of measurement subjective age? The source you cite is Stephens [20],  from 1991. What is the reason for choosing this source and method?

Thank you for your comments. In general, subjective age is often measured by a single question with sensory age ("How old do you feel?"). Kleinspehn-Ammerlahn et al. (2008) added the appearance age ("How old do you feel when you look at yourself in a mirror?"). In this study, a total of four subjective ages were examined together by adding Stephens's (1991) interest and activity age to reflect a scale that can measure subjective age more multidimensionally. Furthermore, the difference score between chronological and subjective age was used. We acknowledge that using scores based on differences can be controversial (Rogosa, 1995). However, when using difference scores, one can directly determine how much younger or older an individual perceives himself (e.g., 3 years younger). In addition, partial calculation of chronological age from subjective age may lead to the disappearance of age-related variances (Kleinspehn-Ammerlahn et al., 2008). We explained this in the introduction as well. Please see lines 82-92.

 Data Analysis

- Did you checked the data distribution normality?

              Yes, we checked normal distribution with the data.

- Did you control for sociodemographic variables as confounders? Please explain

No, we did not control the sociodemographic variables as confounders. Our main aim of this study is to analyze the different means within the groups, not to analyze covariance. The null hypothesis in an ANOVA test is that the means of several groups are all the same, and the alternative hypothesis is that at least one of the groups' means is different. We pointed out the significance of the careful selection of null and alternative hypotheses.

 Results

 -   More detailed information should be given about the age gap at the beginning – the table 2 is not clear. How and why authors introduce the categories: More than 10 years old, Less than 1-5 years old and others

Thank you for your careful comments. We added note under table 2 to clarify the age categories. This current study was not collected by limiting the age of the elderly. However, as a result of collecting actual data and classifying it by chronological age, the number of super-aged people aged 80 or older was small, and the number of elderly people in their 70s was relatively large. After looking at the data collected about the difference between chronological and subjective ages that are important to this study, setting it as a cutting point in units of 5 years and treating it as a group of extremes (more than 10 years old, less than 11 years old) was needed to not break the rule for one-way ANOVA analysis (the dependent variable is normally distributed). Please see lines 274-279.

<Reference>

The results of a one-way ANOVA can be considered reliable as long as the following assumptions are met:

  • the response variable (the dependent variable) is normally distributed;
  • the samples are independent;
  • the variances of populations are equal.

- Above the Table 2: Note. a, b, and c are not clear. The reader may not understand what groups differ from which groups.

Thank you for your careful comments. To clarify the meaning of the letters, we added the notes below Table 2.  Please see lines 270-272.

 Discussion

 -          What are practical implication of the study? Please include some more information about the mental health promotion for elderly – how to use the results for such aims?

Thank you for your valuable suggestion, we revised the discussion section. Please see lines 337-350, 361-371, and 398-401.

 General suggestion

– the paper should be proofread

We asked an expert who is fluent in English to review the entire manuscript and revise it by reflecting on it.

Round 2

Reviewer 2 Report

The paper has been clearly improved. However, in my opinion, some issues pointed in review, and not included in the revised version, should be pointed in Limitations section (e.g. not controling for sociodemographic cofounders, not every tool in Korean version).

Author Response

2nd Round

Reviewer Report on Manuscript # healthcare-2073410 “Aging and Mental Health: The Evidence of Socioemotional Selectivity Theory Using Subjective Age Gap”

Note: To facilitate our responses to all the comments, we have numbered each comment and provided the reviewer’s excerpts in italics. Our detailed responses follow in regular font with red color.

First of all, we sincerely appreciate the time and efforts you have provided again. Your thoughtful suggestions have helped us substantially improve on our original submission. Please find our detailed response to your comments below.

Reviewer2.

The paper has been clearly improved. However, in my opinion, some issues pointed in review, and not included in the revised version, should be pointed in Limitations section (e.g. not controlling for sociodemographic cofounders, not every tool in Korean version).

Thank you so much for your feedback. We added the contents you mentioned to the limitation section. Please see lines 402-410.
